# Targeting Menin and CD47 to Address Unmet Needs in Acute Myeloid Leukemia

**DOI:** 10.3390/cancers14235906

**Published:** 2022-11-29

**Authors:** Andrew H. Matthews, Keith W. Pratz, Martin P. Carroll

**Affiliations:** 1Department of Medicine, Abramson Cancer Center, Perelman School of Medicine, University of Pennsylvania, Philadelphia, PA 19104, USA; 2Department of Medicine, Perelman School of Medicine, University of Pennsylvania, 715 Biomedical Research Building II/III, 421 Curie Boulevard, Philadelphia, PA 19104, USA

**Keywords:** acute myeloid leukemia (AML), target therapy, KMT2A, menin, CD47

## Abstract

**Simple Summary:**

Despite recent, rapid drug development success for patients with acute myeloid leukemia, distinct molecular and genetic aberrations still confer a poor prognosis. In this review, we explore the preclinical and early clinical development of two promising approaches: disrupting menin signaling leading to cell differentiation or blocking CD47 to unlock the innate immune system. These two approaches may improve treatment for patients with high unmet needs today.

**Abstract:**

After forty years of essentially unchanged treatment in acute myeloid leukemia (AML), innovation over the past five years has been rapid, with nine drug approvals from 2016 to 2021. Increased understanding of the molecular changes and genetic ontology of disease have led to targeting mutations in isocitrate dehydrogenase, FMS-like tyrosine kinase 3 (*FLT3*), B-cell lymphoma 2 and hedgehog pathways. Yet outcomes remain variable; especially in defined molecular and genetic subgroups such as *NPM1* (Nucleophosmin 1) mutations, 11q23/*KMT2A* rearranged and TP53 mutations. Emerging therapies seek to address these unmet needs, and all three of these subgroups have promising new therapeutic approaches. Here, we will discuss the normal biological roles of menin in acute leukemia, notably in *KMT2A* translocations and *NPM1* mutation, as well as current drug development. We will also explore how CD47 inhibition may move immunotherapy into front-line settings and unlock new treatment strategies in TP53 mutated disease. We will then consider how these new therapeutic advances may change the management of AML overall.

## 1. Introduction: Continued Need for New Therapies

Every year there are 20,000 new cases of AML diagnosed in the United States [1] and 119,000–352,000 diagnosed worldwide [2]. Induction therapy achieves complete remission (CR) in 60–80% of cases but median survival remains short (8.5 months), and 2-year and 5-year overall survival (OS) rates are only 32% and 24% [3]. OS varies widely by age, performance status, karyotype and molecular changes. While new drug approvals exist for an ever-increasing number of patient subsets, large groups remain with poor prognosis and limited therapy options.

Molecular subsets such as patients with *FLT3* and *IDH* mutations have therapeutic agents approved for use in AML, but many patients lack these lesions and could benefit from development of novel therapeutic approaches. Several subsets, ranging from the common *NPM1* mutation (~33% of AML) to the rarer lysine methyltransferase 2A (*KMT2A*) (5–10% of AML) rearrangements (formerly known as mixed lineage leukemia, MLL), may have a common sensitivity to disruption of the protein menin as these two subsets share a common, targetable final pathway. *NPM1* mutated and *KMT2A* rearrangements are often mutually exclusive, models of both demonstrate common gene expression patterns, preclinical testing shows activity in both settings and clinical trials are underway to see if both groups may benefit. As reviewed elsewhere in this series, [4] *TP53* mutant AML remains a largely incurable disease. Although *TP53* modulating agents are being studied, [5] new approaches including targeting of CD47, an immunologic “don’t eat me” signal, may provide a way forward for these subtypes of disease. In this review, we will focus on two of the types of therapy not addressed in other articles in this series, Menin inhibition and targeting of CD47 using antibodies.

## 2. Menin Biology: A Short Introduction

Extensive work has been done characterizing the KMT2A epigenetic complex and its large group of interacting proteins [6] and we will only briefly introduce this complex topic. Menin inhibition has emerged as a promising target for patients with KMT2A fusion proteins as well as *NPM1* mutations. 11q23/*KMT2A* rearrangements occur in 5–10% of patients, [7] are associated with decreased survival [8,9] and may share a common final pathway with *NPM1* mutated AML [10]. *NPM1* mutation occurs in approximately 35% of all cases and up to 60% of patients with normal karyotype AML [7,11] and prognostic implications are mediated by presence of other mutations including *FLT3*-internal tandem duplication (*FLT3-ITD*). *NPM1* and *FLT3* mutations are now routinely screened at diagnosis with *NPM1* mutation and wildtype *FLT3* carrying a favorable prognosis [12] and *NPM1* mutation with *FLT-3 ITD* carrying a worse prognosis. Patients with both a *FLT3-ITD* and *NPM1* mutation have poorer prognosis and may undergo transplantation in first remission when possible [13]. Development of menin inhibitors has focused on all of these key subsets, with early clinical trial results expected in the near-term.

## 3. Menin Inhibition

Menin, a nuclear protein comprised of 610 amino acid residues, is ubiquitously expressed in various organs. Menin helps regulate tissue-specific gene expression via interaction with chromatin regulators and transcription factors. Chromatin immunoprecipitation studies show menin interacts with thousands of human gene promoters. By linking transcription-factor function to histone-modification pathways, menin can act as a global regulator of transcription with different functions in different tissues [14,15]. Intriguingly, menin suppresses tumorigenesis in endocrine organs, with its loss resulting in both the hereditary *MEN1* syndrome and sporadic parathyroid adenomas, insulinomas, gastrinomas and lung carcinoid tumors [16]. However, menin can also be an essential leukemogenic cofactor for KMT2A fusion proteins [17]. Endogenous *KMT2A*, is the human homolog of the drosophila gene *trithorax* which is essential for murine mid-gestational development [18]. The wildtype KMT2A protein assembles into a protein complex, that in conjunction with transcription factors, marks gene promoters and regulates gene expression. KMT2A protein is a member of a large multiprotein complex that interacts with chromatin and normally acts as a histone H3 lysine 4 (H3K4) methyltransferase, a function critical for epigenetic modification and transcription regulation [19,20]. Both wild type [21,22,23] and mutant forms of the KMT2A protein [24,25] play a role in regulating stem cell phenotype through regulation of homeobox (*HOX*) genes. Chromosomal translocations frequently disrupt transcription factors and lead to leukemogenesis. These translocations can form fusion proteins with novel functions such as new binding partners or changes in interactions with prior binding partners. These changes impact epigenetic regulation and affect hematopoietic differentiation. Most *KMT2A* rearrangements are balanced translocations resulting in the production of in-frame gain-of-function fusion oncoproteins. Importantly, these chromosomal rearrangements alone may be sufficient to generate a full leukemic phenotype, [26] as seen with KMT2A fusion proteins in murine models [27,28,29]. *KMT2A* chromosomal alterations have been extensively studied in mixed lineage leukemia for more than twenty years [30]. *KMT2A* encoded wildtype (wt)-KMT2A requires interaction with menin to maintain *HOX* expression and self-renewal properties of hematopoietic progenitors [31,32,33,34]. Mechanistic mouse models show KMT2A fusion proteins induce leukemic transformation from hematopoietic stem cells in a dose-dependent manner [35,36,37]. As epigenetic regulation becomes dysfunctional a stereotypical pattern of novel target gene expression emerges including overexpression of *FLT3*, [38,39] *PBX3*, [40] *HOXA9* and *MEIS1* [21,41]. Transcription of the latter two genes promotes self-renewal through stem-cell like transcription programs [29,37,42] and is critically dependent on the *KMT2A*-fusion protein binding to menin [17,43].

Balanced translocations that initiate AML, like *KMT2A*r—or *CBFB*-*MYH11*, *RUNX1T1* and *PML*-*RARA*—are often mutually exclusive with *NPM1* mutations [44,45]. Patients with *NPM1* mutated AML commonly have co-mutated *DNMT3A* and/or *FLT3-ITD* and are associated with a characteristic gene expression profile including downregulation of CD34 and overexpression of *HOXA* cluster genes [46]. As noted above this same pattern of *HOXA 7/9/10* and *MEIS1* gene overexpression is also seen in *KMT2A* rearrangements. The Armstrong lab demonstrated that *NPM1* mutant leukemia is dependent on menin binding to wild type KMT2A and loss of KMT2A-menin abrogates the leukemogenic effect of *NPM1* mutations [47]. These observations raise the possibility of disrupting KMT2A protein interactions in both patients with *KMT2A* fusion proteins and those with *NPM1* mutations. There may be a common, shared, sufficient AML pathway vulnerable to menin disruption.

## 4. Targeting Menin in Acute Leukemia

Small molecules disrupting KMT2A and menin interactions show substantial antileukemic efficacy in cellular and murine models of *KMT2A*-rearranged (*KMT2A*r) AML [17,48]. The KMT2A fusion complex is a large multi-protein complex and disruption of the menin-KMT2A fusion protein complex restores histone methylation, gene expression and permits leukemic cell differentiation. Inhibitors thus lead to improved survival in genetically engineered mouse and xenotransplantation models [48,49,50]. Small molecule inhibitors produce the same effects as deleting menin in *KMT2A*r models using a CRISPR/cas9 model system [47]. Menin inhibitors also appear to lead to decreased expression of *HOXa/b* targets, including *MEIS1*, and lead to differentiation in *NPM1/DNMT3a* mutant knock-in mice and leukemia cell lines [51,52]. Based on these promising preclinical data, development has moved into clinical trials focusing on *KMT2A*r and *NPM1* mutated leukemias.

Although many of the details of the KMT2A complex have been understood for some time, developing small molecule inhibitors of this epigenetic complex has progressed slowly [53,54,55]. The focus has now turned to disrupting menin binding to either *KMT2A*r fusion proteins, the wildtype complex or both (Figure 1). After years of optimization and drug selection, four drugs that inhibit MLL-menin binding have recently entered clinical testing as single agents for AML (K0539 (NCT04067336); SNDX-5613 (NCT04065399); JNJ-75276617 (NCT04811560); DS-1594b (NCT04752163); BMF-219 (NCT05153330). After safety and dose optimization are established, efficacy testing will follow in patients with *KMT2A*-rearrangements or *NPM1* mutations.

Preliminary data from the phase I/II KOMET-001 trial (NCT04067336) has been presented. Among patients in four dosing cohorts, median age was 67 years-old, a median of 3 prior lines of therapy had been received and 2 of 12 achieved a complete response (one patient had *SET2D/RUNX1* and the other *NPM1* and *KMT2D* mutations). There were no discontinuations due to adverse events and grade 3 or higher adverse events included elevated lipase, pancreatitis, neutropenia, tumor lysis syndrome and deep vein thrombosis each occurring in 1 patient each (8.3%) while grade 1 or 2 adverse events included nausea, rash and diarrhea [56,57]. The phase I/II AUGMENT-101 trial of SNDX-5613 has preliminary results presented in abstract form from 45 evaluable patients showed an objective response rate of 44%. In the 35 patient subgroup with MLL rearrangement, the ORR was 49% and in the 10 patients with *NPM1* mutation the ORR was 30% (of those with a CR, 70% achieved minimal residual disease negative state). Patients lacking either *KMT2A*r or *NPM1* mutations did not respond [58,59]. There were no treatment discontinuations due to adverse events, however 13% (n = 7) of patients experienced grade 3 QTc prolongation. These trials and others are ongoing (Table 1) but the preliminary conclusion is that these drugs will be tolerated, although QTc monitoring will be important, and they potentially have single agent activity in patients with MLL-rearranged or NPM1 mutant AML.

## 5. Future Steps in Development

Depending on monotherapy activity and tolerability, combination therapy is likely to follow. Early preclinical data favor combinations targeting *FLT3-ITD*. Recent data suggests *NPM1* and *KMT2A* translocations might share significant cooperativity with *FLT3* in leukemogenesis. In particular *MEIS1* is a critical upstream regulator of *FLT3*, [60] therefore disrupting menin may decrease MEIS1 protein expression and FLT3 protein activity. In fact menin inhibition led to a profound downregulation of *FLT3* expression in AML cell lines, [47] opening up the possibility of combination therapy in *NPM1* mutated, *FLT3-ITD* mutated AML. Additionally menin complex inhibitors downregulated and dephosphorylated FLT3, FLT3-ITD inhibitor and menin inhibitor combination therapy led to a left shift of the 50% inhibitory concentration for combination components over the individual drugs, and a murine model of KMT2A-rearranged *FLT3-ITD* mutated AML showed improved survival [61,62]. Other preclinical evidence has demonstrated potential synergy by disrupting menin and inhibiting BCL-2 or CDK6 with venetoclax or abemaciclib, respectively, in both AML cell lines as well as patient derived AML cell xenografts [63].

Competing agents are also targeting the KMT2A machinery and downstream pathways. Numerous protein complexes act as binding partners for KMT2Ar associated fusion proteins and several including PAFc, DOT1L and SEC have all been therapeutic targets [64]. Success has yet to be realized. In recent years, pinometostat, an inhibitor of DOT1L failed to show more than modest activity with two complete responses among 51 patients [65]. Still other groups are targeting the presumed shared downstream pathway components, such as spleen tyrosine kinase (SYK) [66]. Entospletinib, an oral SYK inhibitor, has completed phase Ib/II trial (NCT02343939) and showed favorable response rates when combined with induction chemotherapy in AML patients with *KMT2A* translocations, *NPM1* mutations and *FLT-3 ITD* mutations all with a shared *HOXA9/MEIS1* signature [67]. The planned phase 3 trial will be targeting patients with newly diagnosed *NPM1* mutated AML and may pursue accelerated approval on the basis of measurable residual disease negative complete response as the primary endpoint. Similar to menin inhibitors, SYK inhibitors are being examined in combination with FLT3-ITD inhibitors as well as venetoclax and azacitidine in patients with newly diagnosed AML with *NPM1* and *FLT3* mutations. Significant new work has defined the various KMT2Ar fusion protein containing complexes and their discrete interacting proteins; preclinical work suggests that combinatorial inhibition of menin interactions with, for example, ENL inhibition, may have cooperative or synergistic effects on leukemic cell differentiation. This exciting area of research has been recently reviewed [68].

Overall, menin inhibitors represent a promising target for *KMT2A* translocated and *NPM1* mutated AML, perhaps especially patients with *NPM1* mutated *FLT3-ITD* high AML. These subgroups represent an ongoing unmet need despite several new agents for AML. The near-term results from phase I/II monotherapy trials will elucidate future development. Combination therapy with either traditional chemotherapy, FLT3 inhibitors or other epigenetic modifiers may be an exciting path forward as our understanding of the basic biology of leukemic transformation improves.

## 6. Anti-CD47 Antibodies

Immunotherapy has changed the clinical landscape of solid oncology and is reshaping lymphoma and multiple myeloma treatment. Unfortunately, PD-1 and PD-L1 inhibitor immunotherapies have not been transportable to relapsed/refractory AML [69,70]. Numerous groups are trying to advance therapy earlier in the disease process or combine immunotherapy with hypomethylating agents although early studies have been mixed [71]. While other immunotherapies beyond checkpoint inhibitors have garnered success in lymphoma, myeloma, and even acute lymphoblastic leukemia, progress has stalled in AML. Monoclonal antibodies, T cell engager antibodies, alloreactive NK and CART cell therapy, immune checkpoint blockade beyond PD-1/PD-L1 or CTLA4 such as TIM3 are still actively being investigated [72]. Cellular immunotherapies like autologous NK and CART cells, have thus far been of limited relevance to AML. Even when a target exists, all immunotherapy approaches seem to face hurdles of collateral hematopoietic damage or inactivity. While standard immune checkpoint inhibitors or cell therapies have not yet been successful in AML, strategies to unlock the innate immune response and may represent a much needed new strategy against poor risk AML such as those with *TP53* mutations.

Blocking cluster of differentiation (CD) 47 may provide a path forward. CD47 is a nearly ubiquitous expressed cell surface protein belonging to the immunoglobulin family. It is a heavily glycosylated 50Kd cell surface protein with an extracellular *N*-terminal IgV domain, five transmembrane domains, and a short C-terminal cytoplasmic tail. CD47 is the ligand for signal-regulatory protein (SIRP)α, an inhibitory receptor expressed on myeloid cells. SIRPα is a transmembrane protein whose cytoplasmic domain contains immunoreceptor tyrosine-based inhibition motifs which when phosphorylated recruit phosphatases and inhibits the activity of non-muscle myosin IIa. This decreases the ability of prophagocytic signals on tumor cells to trigger phagocytosis by macrophages [73]. As a cell surface ligand CD47 is present in low levels on nearly all cells [74]. For example, it is a marker of self on red blood cells [75] and expression decreases with age leading to splenic clearance of senescent red blood cells [76]. By interacting with SIRPα on macrophages and dendritic cells it sends a potent “don’t eat me” anti-phagocytosis message signal. This signal serves a role in normal immune tolerance, autoimmunity and both solid [77,78,79,80] and hematological malignancy [81,82,83,84]. It also acts as immune checkpoint with overexpression of CD47 usually contributing to innate and adaptive immune escape [85,86,87,88]. Typically upon cell damage or stress, cells will increase expression of prophagocytic signals, such as calreticulin, that leads to cell removal [89,90]. These signals counteract the phagocytosis checkpoint from CD47-SIRPα axis.

Preclinical studies have shown that CD47 blockade is not sufficient for phagocytosis of tumor cells [91]. For anti-tumor activity, another co-stimulatory signal is needed. Tumor cells may need to express additional pro-phagocytic signals such as SLAMF7 with Mac-1 [92] and calreticulin induced by azacitidine, [93] or additional activation of macrophages via Fcγ receptor activation by the Fc region of a monoclonal antibody targeting CD47 or by a second agent targeting leukemia cells (e.g., CD33). Preclinical findings have also shown that F(ab’)2 fragments of a monoclonal antibody against CD47 showed no apoptotic effect on CD34+ hematopoietic progenitor/stem or human endothelial cells in leukemia xenograft mouse models [94]. Thus the selectivity of CD47 antibodies relies on the balance of CD47 as well as prophagocytic signals on normal cells and leukemia cells (Figure 2).

## 7. Targeting CD47 in Acute Myeloid Leukemia

CD47 is highly expressed in AML cells including leukemic stem cells [95] with approximately 25–30% of patients across multiple AML cohorts having high levels of CD47 mRNA expression in peripheral blood blasts [96]. Interestingly, CD47 mRNA expression varies across cytogenic and molecular subgroups with lower expression in cases with t(8;21), a favorable risk translocation, and higher expression in cases harboring *FLT3-ITD* mutations. High CD47 expression has been shown to be an independent prognostic factor for poor overall survival in AML patient cohorts [96]. Thus CD47 expression is thought to have both prognostic and therapeutic implications in AML.

Blocking CD47 may be most effective in certain subsets of AML, where the balance of pro-phagocytic and anti-phagocytic signals is altered. For example, *TP53* mutant patients have been shown to have an immune anergic profile, immunosuppressive tumor microenvironment and PD-L1 is upregulated on leukemic stem cells in *TP53* mutant AML and MDS [97,98]. Furthermore, p53-null tumors appear able to reeducate myeloid cells such that they attenuate T-cell responses with decreased CD4+ and CD8+ T cell proliferation [99]. *TP53* mutational status may also be critical within the microenvironment itself. In p53 null mice, the size of tumor xenografts are greatly increased and correlate with the presence of myeloid derived suppressor cells, regulatory T cells, and a loss of effector function [100,101]. CD47 blockade may therefore be able to exploit how *TP53* mutations reshape the local innate and adaptive immune response.

Multiple groups established monoclonal antibodies against CD47 that enabled phagocytosis of AML leukemic stem cells in vitro and inhibited their growth in leukemia cell lines and mice models [95,96]. Thereafter, a humanized IgG4 anti-CD47 antibody (magrolimab) demonstrated leukemic eradication and long term survival in xenografted mice [102]. Agents quickly moved into clinical trial settings in multiple malignancies. The first agent targeting CD47 entered clinical trials in 2014, there are now over 20 agents in active clinical development in over 30 trials spanning multiple indications and cancer types (Table 2 lists clinical trial agents in AML). Groups have also begun to target CD47 with recombinant fusion proteins (e.g., TTI-662 a fully human recombinant SIRPαFc fusion protein) [103] and bispecific antibodies [104].

Magrolimab, as the first-in-class clinical trial candidate, has provided early lessons. In a Phase 1 monotherapy trial, magrolimab failed to show any CR or CR with incomplete count recovery (CRi) in patients with relapsed and/or refractory AML [105]. Another early lead agent CC-90002 showed a similar low single agent activity, 36% of patients developed anti-drug antibodies and adverse events included disseminated intravascular coagulation, purpura, heart failure and acute respiratory failure presumably from hemagglutination [107]. The focus has shifted to combination therapies earlier in the treatment course and selecting candidates with low preclinical signals of hemagglutination or hemolysis.

Magrolimab has since shown promising overall response rates in combination with azacitidine for AML and MDS. In a phase Ib trial of untreated AML, among 34 evaluable patients overall response rate was 65% (44% CR, 12% CRi, 3% partial response and 6% morphological leukemia free state, 32% with stable disease. In an interim analysis of 72 patient with *TP53* mutated AML CR was 33% and CRi was 8% with median time to CR of 3 months. 45% (14/31) of evaluable patients with any level of response achieved negative MRD by flow cytometry [108]. As the response of *TP53* AML to conventional treatment is poor, these clinical trial results have lead some investigators to focus anti-CD47 trials on patients with *TP53* mutant AML. Whether TP53 mutation status will be a robust biomarker remains to be determined. Magrolimab has since moved into phase III trials. Other CD47 directed agents are not as far along in clinical development but reflect strategies to either enhance immune activity by coupling CD47 blockade with an activating signal or decrease the risk of hemolysis and hemagglutination.

Toxicity of these agents may continue to present challenges but we are learning more about managing such toxicity. Adverse effects could include hemagglutination and off-target myelosuppression, given the role of CD47 in preventing programmed red cell removal [75]. In the phase 1b magrolimab trial, although anemia was seen in 38% of patients, the anemia predominantly occurred in cycle 1 with a return to baseline by cycle 2 and improvement over time associated with hematologic response. There was a decline in Hb levels in all patients (median Hb change, −1.0 g/dL); eighteen developed a newly positive direct antiglobulin tests without evidence of hemolysis and there were increased transfusion requirements among nineteen patients with relapsed/refractory AML [109]. This has led to so called RBC pruning strategies [110] which has been implemented in AML trials with a magrolimab priming dose-escalation regimen (escalating 1–30 mg/kg given intravenously weekly followed by 30 mg/kg every 2-week dosing in cycle 3 and beyond) to try to mitigate on-target anemia [105].

## 8. Future Development

Assuming hemolysis and hemagglutination are not limiting, numerous potential combinations could further boost efficacy. Although the PD-(L)1 pathway has been disappointing in late-stage AML perhaps combination therapy could be feasible. Preclinical evidence is strongest for the combination of azacitidine and anti-CD-47 antibodies, with the combination leading to higher phagocytosis of AML cells in vitro, induction of proposed pro-phagocytic signals including increased calreticulin, long-term remission in AML xenograft mice models [111]. There are also compelling theoretical combinations to further activate the immune system, such as TIM-3 or CD200 targeting combinations, or to increase the longevity of cellular therapy. While there may be strong mechanistic rationale for other preclinical combinations, such as combing PD-(L)1 axis therapeutics or chimeric antigen T cell receptor therapies, they have yet to be robustly investigated and reported.

Clinical results may very well start to return before robust preclinical evidence. Given *TP53*-mutated AML’s altered immune profile, spanning both innate and adaptive immunity, a trial of atezolizumab, an anti-PD-L1 antibody, and magrolimab will assess activity in relapsed or refractory AML patients (NCT03922477). Additionally CD-47 targeting antibodies could be combined with other monoclonal antibodies such as gemtuzumab ozogamicin. Alternatively, combination with targeted therapy might be feasible if toxicities are not overlapping. Although VIALE-A showed no statistically significant survival benefit for venetoclax and azacitadine versus azacitidine in *TP53* mutated patients, [112] synergism of CD47 blockade and Bcl-2 targeting therapy may be possible as seen in in chronic lymphocytic leukemia [113].

CD47 blockade remains a promising immune checkpoint in AML, and therapies may offer new approaches for hard-to-treat patient populations like those with *TP53* mutations. Magrolimab in combination with azacitidine has provided proof of concept, it remains to be seen if mutational profiling will be sufficient to identify patients who will respond to therapy or if immunological biomarkers will have to be incorporated into trials. Additionally, novel combinations have entered clinical trials and may further increase efficacy. Ongoing large trials will determine if early signals of hemolysis and hemagglutination can be overcome with dose titration or if instead earlier pipeline agents will be needed to overcome this hurdle.

## 9. Conclusions

As noted above and in other articles in this series, AML therapy has advanced significantly in the last few years but remains challenging. Ongoing basic studies continue to both elucidate the basic biology of leukemic transformation and the immunology of the immune escape of AML cells leading to more targeted and less toxic approaches to AML therapy. The days of high dose induction or no therapy are thankfully behind us. Targeting CD47 and menin inhibitors, if successful clinically, would have a major impact on subsets of AML patients with poor outcomes. Menin disruption represents more than twenty years of preclinical work and appears to have the strongest rationale in *KMT2A* rearranged AML but may hold additional potential in *NPM1* mutated AML and even *FLT3-ITD* patients. Early clinical trials have confirmed an efficacy signal even as a monotherapy and beyond QTc prolongation no other major safety signals have yet emerged. On the other hand, CD47 blockade may finally bring immunotherapy to the AML treatment landscape. To date, this strategy appears to be meeting the major challenge of targeting a marker restricted to leukemia cells. Larger trials will attempt to confirm the promise of improved combination therapy efficacy while sparing hematopoietic cells and erythrocytes. These two treatment approaches may reshape management of intermediate and high-risk AML.

## Figures and Tables

**Figure 1 cancers-14-05906-f001:**
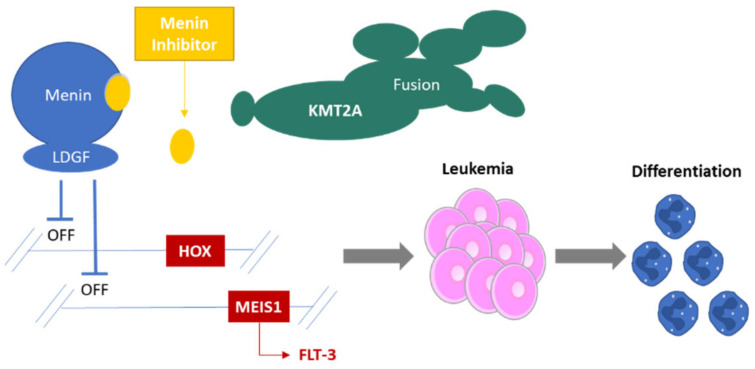
Proposed Mechanisms of Inhibiting Menin. *KMT2A*r fusion proteins bind menin and lead to a stereotyped gene profile expression which includes *HOX* family genes and *MEIS1* leading to a leukemia stem cell like state. The menin inhibitors disrupt binding to menin reducing this gene expression and leading to cell differentiation. *FLT3* expression is regulated by upstream factors including MEIS1. Mutated *NPM1* also up-regulates the expression of *HOXA9* and *MEIS1* possibly by cytoplasmic sequestration of transcriptional regulators (such as HEXIM1) which normally inhibit components of *KMT2A*r fusion protein complexes.

**Figure 2 cancers-14-05906-f002:**
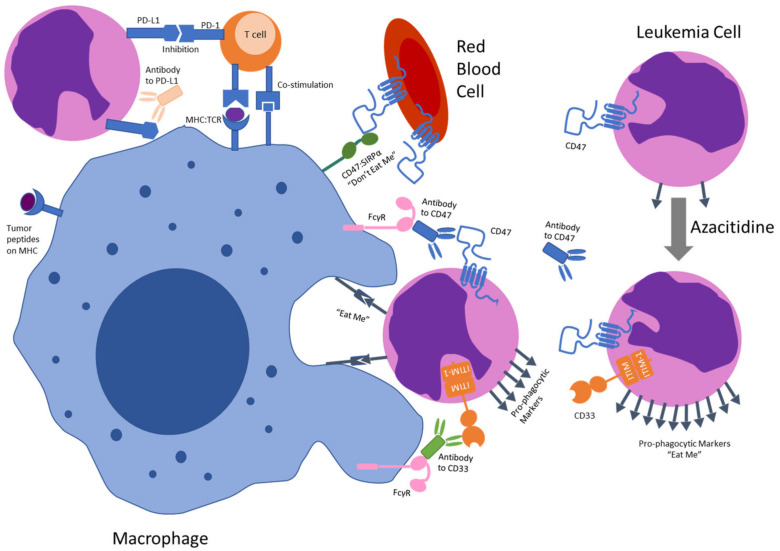
Proposed Mechanisms of Targeting CD47. CD47 blockade decreases the anti-phagocytic “don’t-eat-me” signal on leukemia cells. Azacitidine may upregulate pro-phagocytic signals, such as calreticulin, increasing the likelihood of phagocytosis of leukemia cells. This may be further increased if CD47 binding to SIRPα is blocked or an antibody targeting a leukemic antigen (e.g., CD33) is present to bind a macrophage Fcγ receptor. Phagocytosis may lead to tumor peptide presentation to the adoptive immune system, multiple targets exist to increase T-cell activation such as antibodies to PD-L1 or to PD-1.

**Table 1 cancers-14-05906-t001:** Menin Inhibitors in Clinical Trials. Note: Aza, azacitidine; KMT2Ar histone-lysine-N-methyltransferase 2 rearranged; mini-HCV lower intensity cyclophosphamide, vincristine, doxorubicin and dexamethasone; R/R, relapsed or refractory; ven, venetoclax.

Molecule	Phase	Enrollment	Status	Regimen	NCT	Population
*SNDX-5613*	I/II	186	Recruiting	Monotherapy	NCT04065399	Relapsed/refractory (R/R) MPAL, KMT2Ar AML, *NPM1*m AML
*KO-539*	I/II	100	Recruiting	Monotherapy	NCT04067336	Relapsed/refractory all patients expansion: KMT2Ar and *NPM1*m R/R AML [57]
*JNJ-75276617*	I	110	Recruiting	Monotherapy	NCT04811560	Relapsed/refractory: KMT2Ar, *NPM1m* AML
*BMF-219*	I/II	100	Recruiting	Monotherapy	NCT05153330	Relapsed/refractory all patients (with and without strong CYP3A4 inhibitors)
*DS-1594b*	I/II	122	Recruiting	Combination (ven/aza or Mini-HCV)	NCT04752163	Monotherapy in relapsed/Refractory KMT2Ar AML Monotherapy in relapsed/Refractory *NPM1*m AML Aza/Ven Combo in R/R *NPM1* or KMT2Ar AML Mini-HCVD in R/R ALL with KMT2Ar

**Table 2 cancers-14-05906-t002:** Anti-CD47 Agents in Clinical Development. Abbreviations: AML, acute myeloid leukemia; aza, azacitidine; CMML, chronic myelomonocytic leukemia; F, fit for intensive chemotherapy; MDS, myelodysplastic syndrome; R/R relapsed or refractory; TN, treatment naïve; U, unfit for intensive chemotherapy; Ven, venetoclax. Note: * SIRPa targeting agent, all others target CD47. The following agents are in clinical development but AML patients not included in study: TG-1801, A0-176, TG1801 (anti-CD19 CD47 bispecific), IMC-002, STI-6643, SL-172154, ZL-1201, IMM01, & SRF231 (no longer in active development); PF-07257876 and IBI322 (anti-CD47 and PD-L1); IMM-0306 CD47xCD20 bispecific. SRF231 showed 90% receptor occupancy no complete or partial responses in solid tumor and lymphoma cohorts, development has been suspended. The following agents target SIRPa but AML patients are not included in study: GS-0189, CC-95251, BI765063/OSE-172.

Agent	Phase	Enrollment	Status	Regimen	NCT	Population
*Magrolimab (Hu5F9-G4)*	III	520	Recruiting	Combination with aza	NCT04313881	Treatment naïve (TN) higher risk MDS
*Magrolimab (Hu5F9-G4)*	III	346	Recruiting	Combination with aza	NCT04778397	*TP53* mutated TN unfit (TNU) AML; *TP53* mutated TN fit AML
*Magrolimab (Hu5F9-G4)*	Ib/II	98	Recruiting	Combination with ven/aza	NCT04435691	Ib: Relapsed/Refractory (R/R) AML; II: TNU AML or TN < 75 years old with high risk cytogenetics +/− *TP53*
*Magrolimab (Hu5F9-G4)*	Ib	287	Recruiting [105]	Combination with aza	NCT03248479	R/R AML, TNU AML, RBC transfusion dependent low risk MDS
*Magrolimab (Hu5F9-G4)*	Ib	13	Completed	Combination with Atezolizumab	NCT03922477	R/R AML
*Magrolimab (Hu5F9-G4)*	Ia	20	Completed [106]	Monotherapy	NCT02678338	R/R AML; R/R higher risk MDS
*TJ011133 (TJC4, Lemzoparlimab)*	IIA	80	Recruiting	Combination with aza	NCT04202003	TNU AML or TN higher risk MDS
*TJ011133 (TJC4, Lemzoparlimab)*	Ib	120	Recruiting	Combination with aza or ven/aza	NCT04912063	TNU AML; high risk MDS
*ALX148 (Evorpacept *)*	I/II	97	Recruiting	Combination with ven/aza	NCT04755244	R/R AML or new AML in patients ineligible for standard induction
*AK117*	Ib/II	160	Recruiting	Combination with aza	NCT04980885	AML, newly diagnosed or R/R
*DSP107 **	Ib/II	36	Recruiting	Combination with aza	NCT04937166	TNU AML, TN MDS, R/R MDS/CMML
*TTI-621*	Ia/Ib	260	Recruiting	Monotherapy	NCT02663518	R/R AML
*TTI-662*	Ia/Ib	150	Recruiting	Ia: single agent 1b: combination with ven/aza	NCT03530683	TN *TP53*-mutated AML aza+TTI-662; TNU AML TTI-662 + aza/ven
*IBI188*	Ia	12	Recruiting	Combination with aza	NCT04485065	TN higher risk MDS
*IBI188*	Ib	126	Recruiting	Combination with aza	NCT04485052	R/R AML, TNU AML
*CC-90002*	I	28	Terminated [107]	Monotherapy	NCT02641002	R/R AML or high risk MDS

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
