# Peer review of "Targeting Menin and CD47 to Address Unmet Needs in Acute Myeloid Leukemia"

_cancers, 2022, doi:10.3390/cancers14235906_

Round 1

Reviewer 1 Report

This is a well-written review of the new targeted therapies for relapsed or refractory AML.  The authors focused on menin and CD47 targeted therapies and discussed their clinical implications for refractory AML based on clinical trial results. It is an interesting review and worth to be published in Cancers.

the authors commented on the potential of combination therapies with these targeted agents but did not fully discuss it.

Preclinical studies to investigate the potential and the effect of combination therapies with menin-targeted agents or anti-CD47 antibodies for refractory AML needs to be mentioned.

Author Response

  • Preclinical studies to investigate the potential and the effect of combination therapies with menin-targeted agents or anti-CD47 antibodies for refractory AML needs to be mentioned.
    • Thank you for asking us to expand on this topic. Our prior emphasis on the pre-clinical combination of FLT-3 inhibitors and menin-targeted agents, as well as ENL combinations was incomplete.  We have added content to section 5 paragraph 1 and section 8 paragraph 1 along with additional citations. We also acknowledge in the manuscript that while preclinical evidence for combining azacitidine and CD47 blocking agents  has been demonstrated, there is not yet robust published preclinical evidence for other combinations.

Reviewer 2 Report

In their article, Matthews et al reviews the biological role of some mechanisms in AML bearing some specific genetic abnormalities as NPM1, KMT2A and TP53, discussing the potential impact of some therapeutic options in these settings, also reporting preliminary data from available clinical trials.

The review is indeed comprehensive and well-written, it offers an insightful overview of the treatment scenario in AML in the years to come. I don’t see any major issue.

Minor issues:

Gene names should be reported in italics.

As regards the mentioned FLT3-ITD ratio (paragraph 2), it would be appropriate for sake of clarity to quote the update of ELN 2022, that reconsidered its role in view of the lack of standardization of the method

Some isolated typos should be fixed (for example, section 7 “these results have lead some investigators”; section 9 “would have a major impact subsets”: sounds as one/more words are missing)

Author Response

  • Gene names should be reported in italics.
    • We have corrected this throughout the manuscript.
  • As regards the mentioned FLT3-ITD ratio (paragraph 2), it would be appropriate for sake of clarity to quote the update of ELN 2022, that reconsidered its role in view of the lack of standardization of the method
    • We have updated this paragraph to reflect the ELN 2022 update.
  • Some isolated typos should be fixed (for example, section 7 “these results have lead some investigators”; section 9 “would have a major impact subsets”: sounds as one/more words are missing)
    • Thank you very much for highlighting these issues. These and other typographic errors have been corrected.

Reviewer 3 Report

In this review, the authors described menin and CD47 as targets for treatment of AML. The references are well cited and contents are well written.

I have several minor comments:

1.       Further detailed editing of the manuscript is necessary. There are several minor points for edition, such as

n  Gene names are to be italicized.

n  Gene names are to be written in a uniform format. For example, FLT3-ITD is a more standard writing, but in the manuscript, occasionally other formats are seen. In addition, NPM1 mutation and NPM1c are both used in the manuscript.

n  KMT2A is to be used throughout the manuscript; somewhere MLL is still used.

n  MEIS1 and Meis1 are both visible in the manuscript.

2.       At the end of “Menin Biology: A Short Introduction”, the authors regarded NPM1 mutation and wildtype or low allelic ratio of FLT3-ITD as favorable prognosis. However, in new ELN2022 guideline (Blood (2022) 140 (12): 1345–1377.), FLT3-ITD allelic burden is no longer considered in risk stratification. I suggest the author use this newly published guideline.

3.       Figure 1 is well illustrated, but I suggest HOX and MEIS1 be separated. The current illustration shows HOX followed by MEIS1, and is likely to mislead the readers that these two genes are located in the same chromosome.

4.       At the beginning of “7. Targeting CD47 in Acute Myeloid Leukemia”, the authors described “CD47 is highly expressed in AML cells including leukemia stem cells with approximately 25-30% of patients having high levels of expression”. This sentence is confusing. Please write a clearer statement.

Author Response

  • Further detailed editing of the manuscript is necessary. There are several minor points for edition, such as: Gene names are to be italicized. Gene names are to be written in a uniform format. For example, FLT3-ITD is a more standard writing, but in the manuscript, occasionally other formats are seen. In addition, NPM1 mutation and NPM1c are both used in the manuscript. KMT2A is to be used throughout the manuscript; somewhere MLL is still used.
    • We have updated this formatting and ensured parallelism of gene names.
  • At the end of “Menin Biology: A Short Introduction”, the authors regarded NPM1 mutation and wildtype or low allelic ratio of FLT3-ITD as favorable prognosis. However, in new ELN2022 guideline (Blood(2022) 140 (12): 1345–1377.), FLT3-ITD allelic burden is no longer considered in risk stratification. I suggest the author use this newly published guideline.
    • We have updated this paragraph. We appreciate you and reviewer #2 encouraging us to incorporate the updated ELN guideline.
  • Figure 1 is well illustrated, but I suggest HOX and MEIS1 be separated. The current illustration shows HOX followed by MEIS1, and is likely to mislead the readers that these two genes are located in the same chromosome.
    • Thank you for highlighting this confusing part of our figure. We have updated the figure to clearly show separate chromosomes.
  • At the beginning of “7. Targeting CD47 in Acute Myeloid Leukemia”, the authors described “CD47 is highly expressed in AML cells including leukemia stem cells with approximately 25-30% of patients having high levels of expression”. This sentence is confusing. Please write a clearer statement.
    • We have updated this sentence to better reflect the original manuscript.
  • Journal Formatting
    • We confirm all figures and tables are original.
    • A simple summary has been added to the manuscript.